# Auto-Demo Prompting: Leveraging Generated Outputs as Demonstrations for Enhanced Batch Prompting

## Abstract

Batch prompting is a common technique in large language models (LLMs) used to process multiple inputs simultaneously, aiming to improve computational efficiency. However, as batch sizes increase, performance degradation often occurs due to the model's difficulty in handling lengthy context inputs. Existing methods that attempt to mitigate these issues rely solely on batch data arrangement and majority voting rather than improving the design of the batch prompt itself. In this paper, we address these limitations by proposing "Auto-Demo Prompting," a novel approach that leverages the question-output pairs from earlier questions within a batch as demonstrations for subsequent answer inference. We provide a formal theoretical analysis of how Auto-Demo Prompting functions within the autoregressive generation process of LLMs, illustrating how it utilizes prior outputs to optimize the model's internal representations. Our method effectively bridges the gap between batch prompting and few-shot prompting, enhancing performance with only a slight compromise in token usage. Experimental results across five NLP tasks demonstrate its effectiveness in mitigating performance degradation and occasionally outperforming single prompts. Furthermore, it opens new avenues for applying few-shot learning techniques, such as demonstration selection, within batch prompting, making it a robust solution for real-world applications.

## 1 Introduction

Large language models (LLMs), such as GPT (Brown et al., 2020), and PaLM (Chowdhery et al., 2023), have demonstrated an extraordinary ability to perform in-context learning (ICL), where they utilize provided examples or contextual information to adapt and solve a wide range of downstream tasks. This capability enables LLMs to generalize from few-shot or even zero-shot examples without requiring task-specific fine-tuning, significantly enhancing their versatility across diverse applications (Song et al., 2023). The success of ICL in these models highlights their potential as powerful tools for natural language processing and as adaptable frameworks for learning in dynamic, data-constrained environments, offering broader implications for machine learning and AI research.

Recently, the batch prompting method has attracted growing research interest (Cheng et al., 2023; Lin et al., 2024; Fan et al., 2024), where models are presented with a set of homogeneous questions - queries that share similar structure or content - within a single prompt. The main objective of batch prompting is to enhance interaction efficiency with LLMs by reducing computational costs, especially by minimizing the number of tokens processed. By grouping similar questions, this technique streamlines the model's handling of multiple tasks, optimizing resource usage while ensuring consistent performance across repeated or related queries. As illustrated in Figure 1 (a), the model processes multiple sentences with similar structures in a single batch, identifying and correcting grammatical errors for each input sentence. This demonstrates how batch prompting reduces token usage when working with structurally similar tasks. Moreover, advancements in hardware and algorithmic design have further expanded the capacity of LLMs to retain and process longer input contexts, enabling more effective batch prompting (Munkhdalai et al., 2024; Chen et al., 2023; Ainslie et al., 2023).

```
┌────────────────────────────────────────┐  ┌────────────────────────────────────────┐
│            Batch Prompt                 │  │          Auto-Demo Prompt               │
│                                         │  │                                         │
│ Task Specification: You are a professional  │  │ Task Specification: You are a professional  │
│ NLP expert at sentence grammar check.   │  │ NLP expert at sentence grammar check.   │
│ =========================               │  │ =========================               │
│ Requirement: Generate the answer following  │  │ Requirement: Repeat the input data and generate │
│ the format of the examples below.       │  │ the answer following the format of the examples below. │
│ =========================               │  │ =========================               │
│ [input]                                 │  │ [input]                                 │
│ Input 1: XXXXX                          │  │ Input 1: XXXXX                          │
│ Input 2: XXXXX                          │  │ Input 2: XXXXX                          │
│ ---                                     │  │ ---                                     │
│ [output]                                │  │ [output]                                │
│ Label1: XXXXX                           │  │ {Input 1: XXXXX, Label1: XXXXX}         │
│ Label2: XXXXX                           │  │ {Input 2: XXXXX, Label2: XXXXX}         │
│ =========================               │  │ =========================               │
│ Please generate labels for the [Batch_Size] │  │ Please generate labels for the [Batch_Size] │
│ data given the instructions.            │  │ data given the instructions.            │
│                                         │  │                                         │
│ Input 1: The plan was approved of by my mother. │  │ Input 1: The plan was approved of by my mother. │
│ Input 2: They'll be leaving.            │  │ Input 2: They'll be leaving.            │
│ Input 3: Boston was flown to.           │  │ Input 3: Boston was flown to.           │
│ Input 4: John will have been driving the car. │  │ Input 4: John will have been driving the car. │
│ Input 5: The coat does not fit you.     │  │ Input 5: The coat does not fit you.     │
│ Input 6: Lee never left.                │  │ Input 6: Lee never left.                │
│ Input 7: The roof is leak.              │  │ Input 7: The roof is leak.              │
│ .....                                   │  │ .....                                   │
│ Input 64: Heart disease is considered the leading │  │ Input 64: Heart disease is considered the leading │
│ cause of death in the United States.    │  │ cause of death in the United States.    │
│ =========================               │  │ =========================               │
└────────────────────────────────────────┘  └────────────────────────────────────────┘
```

Figure 1: Example: a) Batch Prompting and b) Auto-Demo Prompting

While batch prompting provides significant advantages in terms of efficiency, it also presents a fundamental challenge, as handling lengthy context inputs is particularly difficult for LLMs based on transformers. Transformer models experience significant performance degradation when accessing relevant information from the middle of long contexts, leading to a decline in overall effectiveness (Liu et al., 2024). This issue becomes more pronounced in tasks that require LLMs to process long contextual inputs. The root of the challenge lies in the quadratic complexity of the self-attention mechanism within transformers, where computational costs increase dramatically with input length. As batch prompting is applied, the combined input length grows, exacerbating this issue and further degrading model performance. Therefore, developing a more effective batch prompting strategy is crucial for mitigating these performance limitations and unlocking the potential of large-scale applications in in-context learning, particularly for tasks involving long sequences of input data.

From the internal representations of LLMs, we observe that batch prompting shares similarities with few-shot prompting, a technique that has proven crucial in enhancing the performance of LLMs by providing a few example demonstrations (Brown et al., 2020). However, an intriguing contradiction emerges: while few-shot prompting typically boosts performance, batch prompting often results in performance degradation. It was found that batch prompting can sometimes outperform single prompts in tasks with smaller batch sizes (Cheng et al., 2023), yet this line of investigation remains underexplored. This raises a key question: Could we bridge the gap between batch prompting and few-shot prompting to leverage the benefits of both?

Interestingly, previous research has found that even when incorrect labels are present in few-shot demonstrations during in-context learning, the decrease in accuracy is generally minimal, typically between 0% and 5% (Min et al., 2022). This suggests that using model-generated answers as demonstrations in few-shot prompts is a viable approach, regardless of the potential for model hallucination or limitations that may lead to incorrect answers.

In this paper, we tackle the performance degradation problem from the perspective of connecting few-shot prompting with batch prompting. **A key insight emerges: the outputs generated during earlier iterations of autoregressive LLMs can, with proper design, be automatically recognized as demonstrations for subsequent text generation, without needing to explicitly include them in the prompt**. Building on this, we propose "Auto-Demo Prompting," a novel batch prompting technique that instructs the model to repeat each question before answering it. The model then au-

tomatically treats the resulting question-answer pairs as demonstrations for the following questions in the batch, eliminating the need to manually pack them into the input prompt.

As illustrated in Figure 1 (b), Auto-Demo Prompting directs the model to produce a sequence of question-answer pairs instead of just answers. This structure ensures that, during the autoregressive generation process, each subsequent question in the batch has access to the generated question-answer pairs from the previous iterations. Consequently, all questions in the auto-demo prompt effectively receive 0 to $N - 1$ additional demonstrations, potentially mitigating the performance degradation associated with long context inputs.

By integrating demonstrations directly into the batch prompting process, our method not only enhances the model's capacity to manage longer inputs but also improves overall task performance, aligning batch prompting more closely with the proven success of few-shot prompting. To assess the effectiveness of Auto-Demo Prompting, we conducted extensive experiments using mainstream models, GPT-4o and GPT-4o-mini, across a variety of downstream tasks. The results demonstrate that the proposed method significantly improves model performance in both an efficient and interpretable way. Our contributions can be summarized as follows:

1. The proposed "Auto-Demo Prompting" represents a pioneering effort in constructing few-shot demonstrations directly within the LLM's inference. While previous research has focused on various external factors associated with batch prompts, our approach uniquely modifies the batch prompt design to enable the model to generate demonstrations as part of its output process.

2. Through extensive comparisons between few-shot prompting and batch prompting, we proposed and validated the hypothesis that Auto-Demo Prompting is approximately equivalent to batch prompting supplemented with few-shot demonstrations. All experimental results showed that Auto-Demo Prompting outperformed traditional batch prompting, supporting our hypothesis and addressing the limitation of performance degradation in batch prompt.

3. We discovered that batch data selection in Auto-Demo Prompting influences the selection of demonstrations (question-answer pairs) during the autoregressive generation steps of inference. By employing a common demonstration selection technique, our results showed significant improvements compared to random batch data selection.

4. The experimental results show that "Auto-Demo Prompting + Batch Data Selection," with batch sizes of 16 and 32, outperforms the single prompts with a batch size of 1. This highlights the considerable potential of integrating additional few-shot prompting techniques into Auto-Demo Prompt to serve as an efficient alternative to traditional single prompts with enhanced accuracy and efficiency.

## 2 AUTO-DEMO PROMPTING

Prompt engineering is essential for unlocking the capabilities of large language models (Marvin et al., 2023). Typically, a standard prompt consists of a task description and a single data point, leading to the development of various prompting techniques to enhance LLM performance on downstream tasks (Sahoo et al., 2024; Besta et al., 2024; Wang et al., 2023; Wei et al., 2022). However, relying on single prompts limits inference to just one data point, making this approach less efficient for large-scale datasets or real-world applications. In contrast, Batch Prompting allows for the processing of multiple data points in a single inference. Cheng et al. (2023) suggests that while batch prompting performs well with smaller batch sizes, there is a tendency for performance to decline as batch sizes grow. Additionally, Lin et al. (2024) noted that varying the order of batch data can yield different results, proposing the integration of these outcomes through majority voting to enhance overall performance.

Unlike previous works, Auto-Demo Prompting aims to maximize the potential of batch prompting and enhance its performance during single inference of large language models. This method guides LLMs to generate a question-answer pair for each input question in the batch, rather than simply providing an answer. By innovating the design of batch prompts with a new output control format, Auto-Demo Prompting ensures that LLMs maintain a consistent output structure across all questions. A key factor in this process is the autoregressive generation mechanism of the decoder in LLMs, which facilitates coherent and contextually relevant output generation.

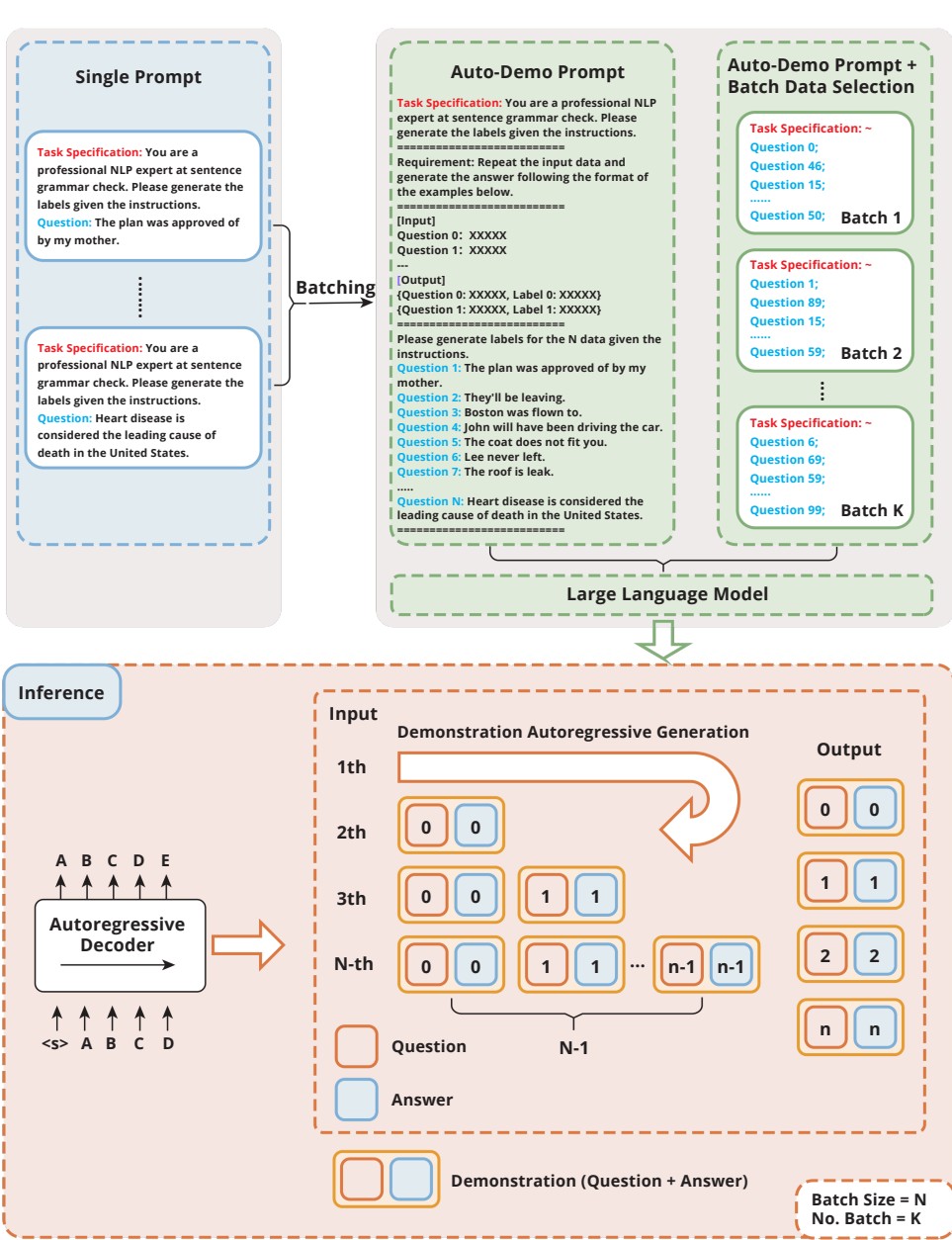

Figure 2: Auto-Demo Prompting: Single prompts are combined into a batch prompt with a special output control for generating question-answer pairs, along with optional batch data selection. This prompt is fed into the autoregressive generation process of a decoder-only LLM, forming demonstrations for subsequent generation.

## 2.1 AUTOREGRESSIVE GENERATION PROCESS IN LARGE LANGUAGE MODEL

Large Language Models can be classified into three categories: encoder-only, encoder-decoder, and decoder-only. The mainstream LLMs are primarily decoder-only and encoder-decoder models. The GPT series developed by OpenAI, including notable examples such as GPT-3, GPT-4, and GPT-4o, consists entirely of decoder-only models and marks significant milestones in the evolution of

generative language models. The Llama series represents open-source LLMs that are also decoder-only. In contrast, examples of encoder-decoder models include T5 (Raffel et al., 2020) and BART (Lewis, 2019). This paper focuses specifically on decoder-only LLMs, where the decoder generates one token at each step of the autoregressive generation process. A single inference of an LLM typically requires multiple steps to produce a complete answer, with the token generated at the previous step added to the current input, serving as the new input for the next token. This iterative process of token generation is known as autoregressive generation.

Based on the generation mechanism of LLMs, we introduce a key insight into the autoregressive generation process: the outputs from earlier steps can serve as prompts or demonstrations for generating subsequent tokens. By employing batch prompting, the model can process multiple input data sequentially, enabling consistent and repetitive responses. This periodicity allows for the use of answers from earlier questions as demonstrations for later ones, thereby enhancing coherence and context retention throughout the generation process.

## 2.2 AUTOREGRESSIVE GENERATION OF DEMONSTRATIONS

As illustrated in Figure 2, Auto-Demo Prompting creatss a loop where demonstrations are generated in an autoregressive manner during the inference of LLMs. This technique guides LLMs to produce question-answer pairs that align with the format of demonstrations used in context learning. Due to the autoregressive nature of token generation, the previously generated output is appended to the current input for the next step, serving as a demonstration for all subsequent questions in the batch. When the batch size is $N$, there will be $\{1, 2, 3, ..., N-1\}$ demonstrations formed in the process.

Suppose the batch size of Auto-Demo Prompting is denoted by $N$. The batch of questions can be represented as $Q = \{q_1, q_2, \ldots, q_N\}$, and the corresponding answers are denoted by $A = \{a_1, a_2, \ldots, a_N\}$. Let $\mathcal{F}$ denote the inference of the LLM. To illustrate the relationships among current batch prompting, Auto-Demo Prompting, and few-shot demonstrations in in-context learning, we compare the formulations of the LLM inference process in these prompting approaches.

---

**Method 1** Few-Shot Prompting

**Formulation:** $\mathcal{F}(a_n | Q_{1:n-1}, \{(q_i, a_i)\}_{i=1}^{n-1}) \quad \forall n \in \{0, 1, \ldots, N\}$
**Example:**
  1: $\mathcal{F}(a_1 | q_1)$
  2: $\mathcal{F}(a_2 | (q_1, a_1))$
  3: $\quad \cdots$
  4: $\mathcal{F}(a_n | (q_1, a_1), (q_2, a_2), \ldots, (q_{n-1}, a_{n-1}))$

---

**Method 2** Batch Prompting (BP)

**Formulation:** $\mathcal{F}(a_n | \text{BP} + Q_{1:n}, \{a_i\}_{i=1}^{n-1}) \quad \forall n \in \{0, 1, \ldots, N\}$
**Example:**
  1: $\mathcal{F}(a_1 | \text{BP} + q_1)$
  2: $\mathcal{F}(a_2 | \text{BP} + q_1 q_2, a_1)$
  3: $\quad \cdots$
  4: $\mathcal{F}(a_n | \text{BP} + q_1 q_2 \ldots q_n, a_1 a_2 \ldots a_{n-1})$

---

**Method 3** Auto-Demo Prompting (ADP)

**Formulation:** $\mathcal{F}((q_n, a_n) | \text{ADP} + Q_{1:n-1}, \{(q_i, a_i)\}_{i=1}^{n-1}) \quad \forall n \in \{0, 1, \ldots, N\}$
**Example:**
  1: $\mathcal{F}((q_1, a_1) | \text{ADP} + q_1)$
  2: $\mathcal{F}((q_2, a_2) | \text{ADP} + q_1 q_2, (q_1, a_1))$
  3: $\quad \cdots$
  4: $\mathcal{F}((q_n, a_n) | \text{ADP} + q_1 q_2 \ldots q_{n-1}, (q_1, a_1), (q_2, a_2), \ldots, (q_{n-1}, a_{n-1}))$

---

Based on the comparison, we illustrate that Auto-Demo Prompting effectively combines conventional batch prompting and few-shot prompting by modifying the output format to enhance in-context learning capabilities. We can express this relationship using an approximate equality:

$$\text{Auto-Demo Prompting} \approx \text{Batch Prompting} + \text{Few-Shot Demonstrations} \tag{1}$$

This approximate equality underscores the significance of Auto-Demo Prompting. As shown in Figure 1 (b), it adds just two additional lines to the Batch Prompt, representing a minor modification. Nevertheless, this small change greatly enhances the autoregressive generation of demonstrations during LLM inference and plays a crucial role in improving the performance of batch prompting.

### 2.3 BATCH DATA SELECTION

In the proposed Auto-Demo Prompting framework, batch data selection - a conventional technique for improving batch prompting performance - can be viewed as demonstration selection. As illustrated in Method 3, the batched data (i.e., questions) is also present within the question-answer pairs generated by the LLM, which serve as demonstrations for subsequent questions. Therefore, we propose the following hypothesis: the selection of batched data in Auto-Demo Prompting achieves a similar effect to the selection of demonstrations in few-shot prompting.

$$\text{Batch Data Selection} \approx \text{Demonstration Selection} \tag{2}$$

Based on this hypothesis, demonstration selection methods can be effectively adapted for batch data selection in Auto-Demo Prompting. Current approaches to demonstration selection are diverse, utilizing criteria such as similarity, mutual information, perplexity, and diversity (Yang et al., 2023). Notably, demonstration selection based on text similarity has proven effective for text pair classification and multiple-choice tasks (Peng et al., 2024; Su et al., 2024).

Inspired by demonstration selection method, we designed the "Batch Data Selection with Retrieval" algorithm to identify similar questions within a single batch. As detailed in Algorithm 1, this algorithm features a data retrieval loop aimed at gathering similar data into one batch. For each batch with size $N$, a target data point is randomly selected. Subsequently, the $N-1$ most similar data points are identified using an embedding model to calculate the pairwise similarity.

---

**Algorithm 1** Batch Data Selection with Retrieval

1: **Input**: batch size = $N$, dataset $D = \{d_1, d_2, ..., d_{|D|}\}$, batched data $BD = \{\}$
2: **Output**: $BD$
3: **for** $d_i \in D$ **do**
4: $\quad temp\_batch = \{d_i\}$
5: $\quad$ **for** $d_j \in D$ s.t $d_j \neq d_i$ **do**
6: $\quad\quad$ Calculate similarity between $d_i$ and $d_j$ using embedding model
7: $\quad\quad$ **if** $d_j$ is among the most similar $(N-1)$ **then**
8: $\quad\quad\quad temp\_batch.add(d_j)$
9: $\quad\quad$ **end if**
10: $\quad$ **end for**
11: $\quad BD.add(temp\_batch)$
12: **end for**

---

## 3 EXPERIMENTS

### 3.1 EXPERIMENTAL SETTINGS

We conducted comparative experiments between Auto-Demo Prompting and conventional batch prompting across various NLP tasks, including question answering (BoolQ), mathematical reasoning (GSM8K, SVAMP), textual entailment (RTE), and paraphrase detection (Quora Question Pairs, QQP). Specifically, the batch data selection experiments were focused on the RTE and QQP datasets to assess the effectiveness of our approach.

### 3.1.1 DATASETS

**BoolQ:** The Boolean Questions (BoolQ) dataset comprises 15,942 yes/no questions derived from real-world queries to the Google search engine (Clark et al., 2019). Each example includes a question, passage, and answer, with an optional page title for context. The dataset is split into 9,427 training examples and 3,270 validation examples, noted for its complexity requiring advanced reasoning and entailment inference.

**RTE:** The Recognizing Textual Entailment (RTE) dataset, compiled from the RTE1, RTE2, RTE3, and RTE5 challenges (Poliak, 2020), includes 2,490 training samples, 277 validation samples, and 3,000 testing samples. It features premise-hypothesis pairs, each labeled to indicate their entailment relationship, making it a valuable resource for advancing natural language inference research.

**GSM8K:** The Grade School Math 8K (GSM8K) dataset consists of 8,500 diverse grade school math word problems designed for multi-step reasoning tasks (Cobbe et al., 2021). It is divided into 7,500 training problems and 1,000 test problems, with each problem requiring 2 to 8 steps to solve. Solutions are provided in a step-by-step format, and the dataset is available in both standard and "Socratic" formats, which include meta-reasoning prompts.

**QQP:** The Quora Question Pairs (QQP) dataset contains over 400,000 question pairs from the Quora platform, each labeled to indicate semantic equivalence. This dataset is essential for tasks like paraphrasing and duplicate question identification, enabling researchers to explore various machine learning techniques for semantic textual similarity (Wang et al., 2017).

**SVAMP:** The SVAMP dataset features 1,000 simple math word problems aimed at assessing NLP models, designed for up to fourth-grade level and involving single-variable equations (Patel et al., 2021). Created to challenge models with variations of existing problems, it tests their contextual understanding and problem-solving accuracy.

### 3.1.2 IMPLEMENTATIONS

We evaluate Auto-Demo Prompt with varying batch sizes over GPT-4o-mini and GPT-4o. The prompts used for the different datasets are provided in Appendix A.

**Embedding Model for Batch Data Selection:** The embedding model utilized is "iic/nlp_corom_sentence-embedding_english-base," available on the ModelScope platform. This model is a dual-tower text representation architecture that employs the CoROM model as its foundation for the pre-trained language model. The training data is derived from the official open-source MS MARCO Passage Ranking dataset (Bajaj et al., 2018).

**Parameters:** All experiments were conducted in an in-context learning setting, without any model training or fine-tuning. The temperature for model inference was consistently set to 0 to ensure uniformity in responses. Batch sizes were set to 1/16/32 for the gpt-4o-mini, which has a maximum output token limit of 16k, and to 8 or 16 for the gpt-4o, with a maximum output token limit of 8k.

### 3.2 RESULTS AND DISCUSSIONS

As shown in Figure 3, the proposed method, highlighted in red, consistently outperforms Batch Prompt in most experiments, allowing LLMs to generate longer outputs without compromising performance. This highlights the beneficial impact of Auto-Demo Prompting, where the generated question-answer pairs enhance the accuracy of subsequent questions within the same batch. Notably, when applied with GPT-4o on the GSM8K dataset, Auto-Demo Prompt achieved an accuracy of 95.7% with a batch size of 16, surpassing the 95.3% accuracy of the single prompt (batch size = 1). A similar trend was observed with the SVAMP dataset, demonstrating the effectiveness of Auto-Demo Prompting in improving the performance of conventional batch prompts.

Our findings align with previous research that highlights few-shot prompting as an effective strategy for enhancing model accuracy, particularly in datasets that necessitate sophisticated reasoning steps (Wei et al., 2022). The proposed method demonstrated significant effectiveness on the BoolQ dataset, which consists of naturally occurring yes/no questions that often require complex reasoning. In contrast, the effectiveness of Auto-Demo Prompt is less pronounced in simpler datasets like

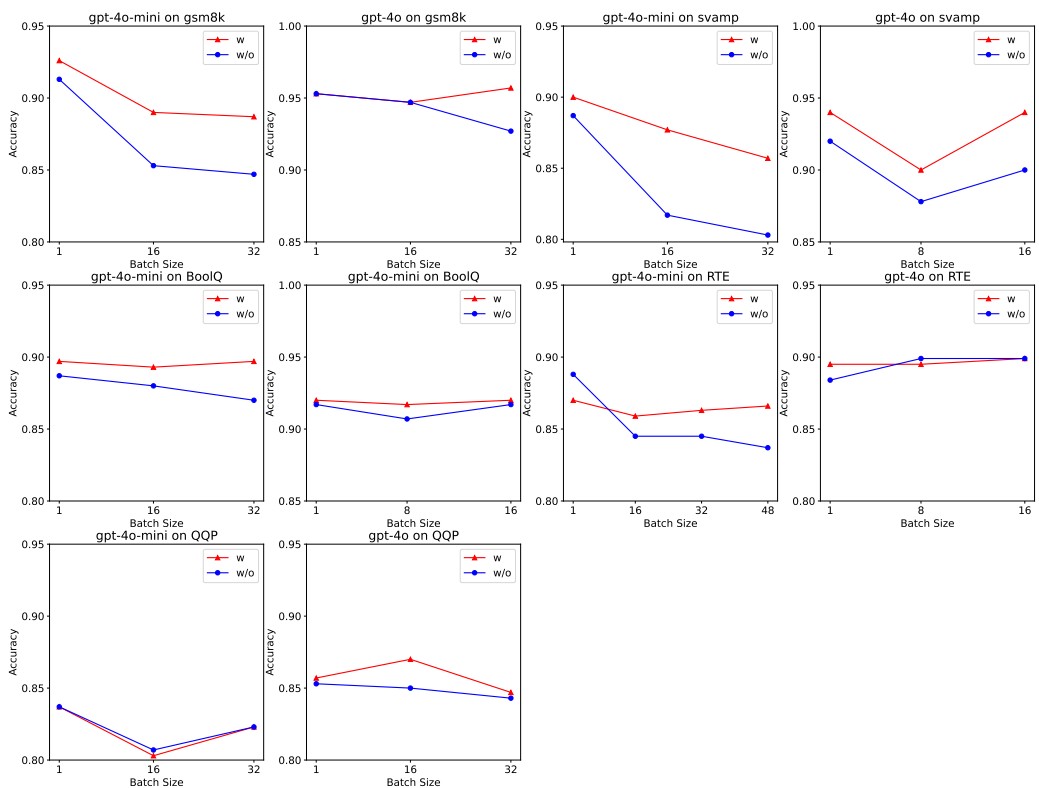

Figure 3: Experimental results: accuracy for Auto-Demo Prompt ("w") vs. Batch Prompt ("w/o") across different models and batch sizes.

RTE and QQP, as the GPT-4o model performs relatively well on these tasks without the need for additional prompting.

### 3.2.1 BATCH DATA SELECTION EXPERIMENTS

To validate the hypothesis stated in Equation 2, we conducted experiments on the RTE and QQP datasets to evaluate the effectiveness of selecting similar questions for batch data selection within the Auto-Demo Prompting framework. As shown in Figure 4, the 'Batch Data Selection with Retrieval' algorithm consistently improves accuracy compared to randomly selected batched data. The results of these experiments closely align with the effectiveness of few-shot demonstration selection based on similarity, as both approaches lead to significant performance improvements.

Surprisingly, the results indicate that experiments utilizing larger batch sizes significantly outperform those employing a single prompt (i.e., batch size 1). For the RTE dataset, the Auto-Demo Prompting approach with a batch size of 48 using the gpt-4o-mini model achieved an accuracy of 89.5%, surpassing the 88.8% accuracy obtained with a single prompt. Similarly, the gpt-4o model with a batch size of 16 achieved an accuracy of 90.3%, exceeding the 88.4% accuracy of the single prompt approach. In the QQP dataset, the batch size of 32 with the gpt-4o model yielded an accuracy of 87.3%, which is 2.0% higher than the result from the single prompt. These findings suggest that the proposed method not only addresses the limitation of performance degradation associated with larger batch sizes, but further enhances overall performance by leveraging previous questions as demonstrations in the autoregressive generation process. This improvement aligns closely with the effects of many-shot learning and underscores the importance of incorporating multiple examples in the prompting process.

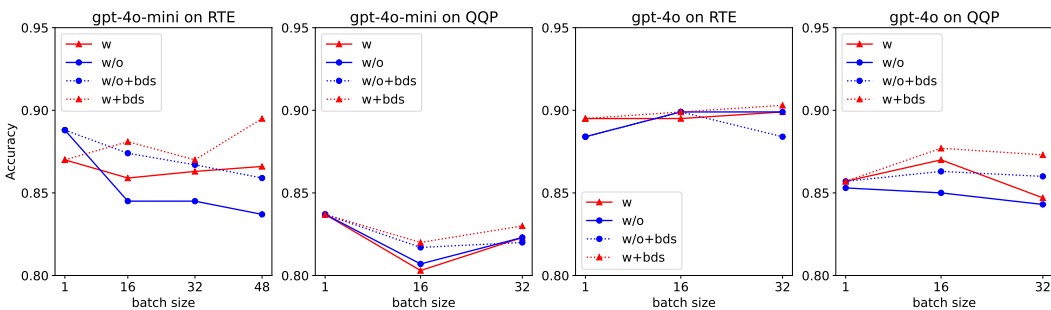

Figure 4: Experimental results for Batch Data Selection: Auto-Demo Prompting ("w") vs. Batch Prompting ("w/o"); "+bds" signifies the application of batch data selection with retrieval.

### 3.2.2 DISCUSSION

The results of our experiments underscore the significant potential of Auto-Demo Prompting. These findings corroborate our earlier observations regarding equations 1 and 2, demonstrating that Auto-Demo Prompting consistently enhances Batch Prompting across most experimental scenarios. It is particularly noteworthy the performance improvement observed when batch data selection is applied. By grouping similar data within each batch, we found that Auto-Demo Prompting with larger batch sizes outperformed configurations utilizing a batch size of one, which is the standard prompt used in mainstream LLM applications. This advancement signifies a promising avenue for achieving both enhanced performance and increased efficiency.

Furthermore, recent research by (Agarwal et al., 2024) demonstrates that many-shot demonstrations can outperform few-shot demonstrations, a trend observed across various domains (Jiang et al., 2024; Moayedpour et al., 2024). As the input and output length limits of LLMs continue to expand, the adoption of a more effective batch data selection method is likely to enhance these performance gains, further solidifying the benefits of our approach. Given the diverse and numerous demonstration selection methods available for in-context learning, the optimal choice can vary across different datasets. This variability encourages future research to evaluate their effectiveness and to develop novel approaches that are better suited for the Auto-Demo Prompting.

## 4 RELATED WORK

### 4.1 BATCH PROMPTING

Batch prompting has emerged as a significant area of research in LLMs to facilitate efficient batch processing of many data points simultaneously. Cheng et al. (2023) first introduced the concept of batch prompting to reduce computational costs, with experiments focusing on batch sizes of fewer than six demonstrating comparable performance to standard prompting. Building on this, Lin et al. (2024) proposed "BatchPrompt," which highlights the variability in results caused by different data orders and introduces a self-consistency method to address these discrepancies. With its cost-saving benefits, batch prompting has been applied in various NLP applications; for instance, Fan et al. (2024) proposed a framework utilizing a covering-based demonstration selection strategy for entity resolution, effectively balancing matching accuracy and computational cost, while Zhang et al. (2023) explored the impact of batch size on several data preprocessing tasks.

### 4.2 DEMONSTRATION SELECTION IN IN-CONTEXT LEARNING

To improve the performance of in-context learning, previous studies have explored the optimization of the selection and arrangement of few-shot demonstrations (Rubin et al., 2022; Zhang et al., 2022; Wu et al., 2023; Fu et al., 2023; Zhou et al., 2023). It was found that the choice of demonstrations is both data-dependent and model-dependent, leading to the proposal of a selection method that considers both factors (Peng et al., 2024). Empirical research by (Min et al., 2022) has demonstrated that preserving the structured format of demonstrations, which consists of text-label pairs, is

essential for optimal performance. They also found that randomly altering the labels within these demonstrations has a negligible effect on performance, providing a foundation for our research.

Our work diverges from previous research in several key aspects. While existing studies have explored batch prompting and its applications, they often overlook the relationship between few-shot prompting and batch prompting, which we believe presents an emerging opportunity for directly enhancing batch prompting. To address this gap, we introduced the Auto-Demo Prompt, which not only improves performance but also offers greater interpretability by establishing a close alignment with the principles of in-context learning and demonstration selection techniques.

## 5 CONCLUSION

In this paper, we introduce the Auto-Demo Prompt, a novel batch prompting method aimed at improving the performance of batch prompting. We provide a comprehensive overview of its operational mechanism and clarify the key concepts that establish the relationship between few-shot demonstrations in in-context learning and batch prompting. Importantly, we show that 'Batch Data Selection' can be conceptualized as 'Demonstration Selection' within the framework of Auto-Demo Prompting, facilitating the transfer of insights from few-shot learning research to our approach. Our extensive experiments validate the effectiveness of Auto-Demo Prompting, underscoring its significance as the input and output lengths of LLMs increase. As these lengths expand, the importance of efficient and high-performing batch prompting methods, such as Auto-Demo Prompting combined with Batch Data Selection, will become increasingly pronounced. Further research is encouraged to investigate demonstration selection in greater detail, as well as to explore the potential synergies between Auto-Demo Prompting and other emerging techniques such as prompt learning and explainable AI. By delving deeper into these areas, we can uncover new insights that may enhance the adaptability and effectiveness of LLMs, ultimately paving the way for advancements in their applications toward achieving artificial general intelligence (AGI).

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

# A  AUTO-DEMO PROMPT DESIGN

Table 1: The auto-demo prompt design for BoolQ dataset

---

**Auto-Demo Prompt for BoolQ Dataset**

---

**Instruction** You are a professional NLP expert at Question Answering annotation. Please generate labels given instructions. You will be given [BATCH-SIZE] passages with questions each time, as input. Each input includes a 'passage' and a 'question' about the passage. Your goal is to determine whether the answer to the question is yes or no and classify, as below:

**Possible Answer:**
[class 0]: if the answer is 'No'
[class 1]: if the answer is 'Yes'.

You will be given [BATCH-SIZE] inputs each time.
============
**Requirement**: Repeat the input data and generate the answer following the format of the examples below.

{Input 1: xxxxx, Label 1: [class X]}
{Input 2: xxxxx, Label 2: [class X]}
============
Please make sure each generated label is in format of [class X].
Please make sure to generate [BATCH-SIZE] labels.

Table 2: The auto-demo prompt design for GSM8K dataset

---

**Auto-Demo Prompt for GSM8K Dataset**

---

**Instruction:** You will be given [BATCH-SIZE] math problems. These problems take between 2 and 8 steps to solve, and solutions primarily involve performing a sequence of elementary calculations using basic arithmetic operations to reach the final answer.

The answer is [numeric result]

You will be given [BATCH-SIZE] inputs each time.
============
**Requirement:** Repeat the input data and generate the calculation results following the format of the examples below.

{Input 1: xxxxx, Reasoning: xxxxx, Answer: The answer is [number]}
{Input 2: xxxxx, Reasoning: xxxxx, Answer: The answer is [number]}
============
Please make sure to write a series of intermediate reasoning steps.
Please ensure the final sentence is "The answer is xxx", where the answer should be a number.
Please make sure to generate [BATCH-SIZE] labels.

---

Table 3: The auto-demo prompt design for QQP dataset

---

**Auto-Demo Prompt for QQP Dataset**

---

**Instruction:** You are a professional NLP expert at duplicate question detection. Your goal is to determine whether two questions are duplicates of each other.

**Possible Answer:**
[class 1]: if they have the same meaning (semantically equivalent).
[class 0]: if they do NOT have the same meaning.

You will be given [BATCH-SIZE] question pairs each time.
============
**Requirement:** Repeat the input data and generate the answer following the format of the examples below.

{Question pair 0: (Question1: xxxxx; Question2: xxxxx), Answer: [class X]}
{Question pair 1: (Question1: xxxxx; Question2: xxxxx), Answer: [class X]}
============
Please make sure each generated label is in the format of [class X].
Please make sure to generate [BATCH-SIZE] labels.

---

Table 4: The auto-demo prompt design for RTE dataset

**Auto-Demo Prompt for RTE Dataset**

---

**Instruction:** You are a professional NLP expert at sentence pair relationship annotation. You will be given [BATCH-SIZE] sentence pairs from the Textual Entailment Recognition dataset each time, as input. Each data includes a sentence pair, "Premise" and "Hypothesis". Your goal is to classify the sentence pair into two classes as below:

**Possible Answer:**
[class 1]: the given Hypothesis and Premise are logical and following (entailment) to each other.
[class 0]: the given Hypothesis and Premise are NOT following (entailment) to each other.

You will be given [BATCH-SIZE] sentence pairs each time.
============
**Requirement:** Repeat the input data and generate the answer following the format of the examples below.

{Sentence pair 0: Premise: xxxxx, Hypothesis: xxxxx, Label: [class X]}
{Sentence pair 1: Premise: xxxxx, Hypothesis: xxxxx, Label: [class X]}
============
Please make sure each generated label is in the format of [class X].
Please make sure to generate [BATCH-SIZE] labels.

Table 5: The auto-demo prompt design for SVAMP dataset

**Auto-Demo Prompt for SVAMP Dataset**

---

**Instruction:** You will be given [BATCH-SIZE] math problems. Each one has a body and a question, please read them and give the equation and answer.

You will be given [BATCH-SIZE] inputs each time.
============
**Requirement:** Repeat the input data and generate the calculation results following the format of the examples below.

{Input 0: Body: xxxx, Question: xxxx, Equation: xxxx, Answer: The answer is [number]}
{Input 1: Body: xxxx, Question: xxxx, Equation: xxxx, Answer: The answer is [number]}
============
Please make sure the final sentence is "The answer is xxx", and the answer should be a number.
Please make sure to generate [BATCH-SIZE] labels each time.

# B EXPERIMENTAL RESULTS

Table 6: Experimental results: "w" denotes the Auto-Batch Prompt, "w/o" denotes the conventional batch prompt, and "bs" refers to the batch size.

| Dataset | Model (Method) | bs=1 | bs=8 | bs=16 | bs=32 | bs=48 |
|---------|----------------|------|------|-------|-------|-------|
| GSM8k | gpt-4o-mini (w) | 0.926 | - | 0.890 | 0.887 | - |
| | gpt-4o-mini (w/o) | 0.913 | - | 0.853 | 0.847 | - |
| | gpt-4o (w) | 0.953 | 0.947 | - | 0.957 | - |
| | gpt-4o (w/o) | 0.953 | 0.947 | - | 0.927 | - |
| SVAMP | gpt-4o-mini (w) | 0.900 | - | 0.877 | 0.857 | - |
| | gpt-4o-mini (w/o) | 0.887 | - | 0.817 | 0.803 | - |
| | gpt-4o (w) | 0.940 | 0.900 | - | 0.940 | - |
| | gpt-4o (w/o) | 0.920 | 0.878 | - | 0.900 | - |
| RTE | gpt-4o-mini (w) | 0.870 | - | 0.859 | 0.863 | 0.866 |
| | gpt-4o-mini (w/o) | 0.888 | - | 0.845 | 0.845 | 0.837 |
| | gpt-4o-mini (Data Selection, w) | - | - | 0.881 | 0.870 | 0.895 |
| | gpt-4o-mini (Data Selection, w/o) | - | - | 0.874 | 0.867 | 0.859 |
| | gpt-4o (w) | 0.895 | 0.895 | - | 0.899 | - |
| | gpt-4o (w/o) | 0.884 | 0.899 | - | 0.899 | - |
| | gpt-4o (Data Selection, w) | - | 0.899 | - | 0.903 | - |
| | gpt-4o (Data Selection, w/o) | - | 0.899 | - | 0.884 | - |
| BoolQ | gpt-4o-mini (w) | 0.897 | - | 0.893 | 0.897 | - |
| | gpt-4o-mini (w/o) | 0.887 | - | 0.880 | 0.870 | - |
| | gpt-4o (w) | 0.920 | 0.917 | - | 0.920 | - |
| | gpt-4o (w/o) | 0.917 | 0.907 | - | 0.917 | - |
| QQP | gpt-4o-mini (w) | 0.837 | - | 0.803 | 0.823 | - |
| | gpt-4o-mini (w/o) | 0.837 | - | 0.807 | 0.823 | - |
| | gpt-4o-mini (Data Selection, w) | - | - | 0.820 | 0.830 | - |
| | gpt-4o-mini (Data Selection, w/o) | - | - | 0.817 | 0.820 | - |
| | gpt-4o (w) | 0.857 | 0.870 | - | 0.847 | - |
| | gpt-4o (w/o) | 0.853 | 0.850 | - | 0.843 | - |
| | gpt-4o (Data Selection, w) | - | - | 0.877 | 0.873 | - |
| | gpt-4o (Data Selection, w/o) | - | - | 0.863 | 0.860 | - |

