# OpenReview forum: "Auto-Demo Prompting: Leveraging Generated Outputs as Demonstrations for Enhanced Batch Prompting"
_ICLR.cc/2025/Conference — Submitted to ICLR 2025_

### Official Review · Reviewer_qraJ · 2024-10-28

**Soundness:** 1
**Presentation:** 1
**Contribution:** 1
**Rating:** 1
**Confidence:** 5

**Summary:**

The paper explores batch prompting, where the prompt groups similar questions to be able to optimize execution efficiency. The work examines how to shape the input context of the prompt to achieve improved quality in the batch prompting scenario.

**Strengths:**

The paper evaluates on multiple datasets
The idea that we can exploit the similarity across questions to reap efficiency improvements is interesting

**Weaknesses:**

The writing is not very clear; there are several unnecessary details. For instance, in the intro:
- The definition of batch prompting is not clear in the intro Lines 43-48; Why does it help efficiency? The writing assumes that the reader is familiar with this.
- How does one decide which questions to batch?
- It is not clear where the model-generated answers as few-shot prompts come into play in Lines 100-102. - - The concept is not introduced in the intro before this.
- What does “proper design” mean on Line 105?
- It is not clear why having 0 to N-1 additional demos would help with long context (Lines 114-115)
- The contributions (Lines 123-141) say nothing about efficiency, despite this being an initial motivation for batch prompting
- It is also not clear why the reader needs any of the details in lines 213-219

How does this compare to methods like Hydragen (https://arxiv.org/abs/2402.05099), which do not increase the context length, but rather share the KV-cache on the shared prefix?

The Batch Data Selection with Retrieval approach is not novel – many approaches select similar examples via these embedding similarity scores

Results:
- The improvements from this method are unclear (0.2%, Lines 370-371)
- There is no comparison to few-shot learning with retrieval-based few-shot selection
- A motivation of batch prompting was efficiency – there is no evaluation of the efficiency improvements

**Questions:**

This work is not up to par for the conference and it is unlikely that author responses will change my opinion

---

> ### Author Response · Authors · 2024-11-18
> **repsonse for reviewer**
>
> Thank you for your reviews.
> The definition of batch prompting is not clear in the intro Lines 43-48; We will definite the detail of "batch prompting" and introduce the advantage of it.
> The efficiency experiments will be added.
>
> **Motivations:**
> Our main idea is to revisit the detail of the autoregressive inference of Batch prompting and amplify the demonstration role of batch data. Previous batch prompting works( BatchPrompt: accomplish more with less, Batch Prompting: Efficient inference..) have claimed that some previous data in batch may play a role of demonstration. However，there are not enough evidence and explanation on this. After we revisit the inference process of batch prompting, we find that the advantage of mutiple data is neglected. Previous works do not make most of the demonstration role of multiple data in batch prompting, which can be used to improve the performance.
>
> **Novelty**
>
> (1) We are the first to construct the squential demonstrations in the output of large language models. And We show its effectiveness by the experiments. Especially, the obivious improvement at GSM8k and SVAMP,  almost %5 higher than “batch Prompting”。
>
>
> (2) Further, we find that "similar batch data" leading "simialr demonstrations in the output" will be quite beneficial to the perfromance on condition with Auto-demo promting.
>
> (3) what we want to argue is that The best strength of Our method is that we get higher accuracy than conventional single prompting with less token cost and lower average time cost per data.  As figure 4 shows, 3 results are better than single data inference with less cost.
>
> ***the question***
> How does this compare to methods like Hydragen (https://arxiv.org/abs/2402.05099), which do not increase the context length, but rather share the KV-cache on the shared prefix?
> They are two kinds of efficient inference techniques. I will list some shortages of prefix cache to prove the value of "batch prompting".
>
>
> (1) Cache Pollution: prefix cache can cause the cache to be filled with a large amount of infrequently accessed data, a phenomenon known as cache pollution. When the cache is filled with irrelevant data, the data that is really needed may be expelled from the cache, thus reducing the cache efficiency.
>
>
> (2) Cache Coherence Issues: In multi-core systems, prefix cache may need to maintain cache coherence, which can add additional complexity and overhead, especially when it is necessary to ensure that multiple cores or processors are seeing the most recent data
>
>
> (3) Performance Variability: The performance of the prefix cache can vary widely from application to application and workload. In some cases, the prefix cache may not provide the expected performance improvement or may even degrade performance due to interaction with the hardware prefetch mechanism
>
>
> (4) Hardware Support Requirements: prefix cache may require specific hardware support, such as Content Addressable Memory (CAM), which can add cost and design complexity
>
>
> (5) Adaptability Issues: The prefix cache may not adapt to the environment that dynamically changes, for example, the routing table changes frequently. This may cause the prefixes in the cache to quickly become outdated, requiring frequent updates
>
> ***response*** We think Auto-demo Prompting can work well with many prefix techniques. Further, Auto-demo prompting do not require some cache room to enhance the efficiency. Obiviously, Batch prompting and AutoDemo Prompting have less hardware requirements.

---

### Official Review · Reviewer_hmYZ · 2024-11-03

**Soundness:** 2
**Presentation:** 2
**Contribution:** 2
**Rating:** 3
**Confidence:** 4

**Summary:**

This paper proposes a method called auto-demo prompting, aiming to improve performance when batched processing is needed in in-context learning. The authors propose to ask the model to repeat each question in the batch following by answering each question, showing that this method can outperform previous batching methods that simply ask the model to provide the answers.

**Strengths:**

- The proposed method is interesting, showing that repeating the question in the batch can better help the model answer each question more accurately.

- The experiments are relatively comprehensive over 5 commonly used NLP datasets.

**Weaknesses:**

- The main idea is rather straightforward, it can be regarded as a simple optimization over the initial prompt used for batching requests. There could be many other possible optimizations to the initial prompt for improved performance (e.g., insert label ids after each input and ask the model to predict the label ids), it would be more interesting if this paper can provide a more comprehensive study rather than proposing a single optimization technique.

- The current analysis in the paper is not sufficient to support the claim that the performance improvement is due to "earlier outputs act as demonstrations of later questions". It could be simply that after repeating each question, the model pays better attention to the question. Can the authors provide more analysis on this to support their main claim?

- In experiments, can the authors add standard deviation by running over multiple permutations of the batched examples? It's unclear if the current results are statistically significant since the numbers are very close.

- For the main results in Figure 3, why the batch sizes (x-axis) are inconsistent? RTE has 1, 16, 32, 48, BoolQ has 1, 8, 16, while other tasks have 1, 16, 32. Can the authors run more comprehensive experiments and show results with batch size 1, 8, 16, 32, 48?

**Questions:**

- Can the authors add standard deviations to Figure 3?

- Can the authors make the x-axis / batch-size range consistent across experiments?

---

### Official Review · Reviewer_4PiS · 2024-11-04

**Soundness:** 1
**Presentation:** 2
**Contribution:** 1
**Rating:** 3
**Confidence:** 4

**Summary:**

In batch prompting, many inputs are formatted consecutively in the prompt, and the model then produces responses for each input in single generation. This can improve computational efficiency by reducing the number of tokens processed (for instance, only needing one instruction for many samples). However, batch prompting tends to degrade prompting performance, while few-shot prompting---which uses the same in-context examples, but formatted differently---can improve it. Motivated by this observation, the paper proposes Auto-Demo prompting, which follows the format of batch prompting but instructs the model to output both the question and the answer, given the question. The intuition is that the model can then treat the consecutive (question, answer) pair as a traditional few-shot demonstration, while still attaining the parallelism of batch prompting. The paper also considers how to better select demonstrations, and finds that the method can outperform standard batch prompting techniques.

**Strengths:**

Quality: auto-demo prompt outperforms standard batch prompting on 7/10 tasks across batch sizes.

Significance: improving inference-time efficiency in terms of prompting techniques is an important aspect of using LLMs on large amounts of data.

**Weaknesses:**

Quality:
- Limited evaluation: I could not find any baseline against few-shot prompting, which auto-demo prompting aims to "combine" with batch prompting. Moreover, there is no evaluation of efficiency/cost, even though I believe auto-demo prompting generates more tokens and thus is more expensive. Given this tradeoff and the lack of evaluation against few-shot prompting, it is unclear if we can conclude that auto-demo prompting provides a cost-performance tradeoff improvement over standard methods. Overall, the performance improvements appear to be minor, while incurring extra cost.
- Missing ablations: what is the role of the embedding space, and do other metrics for data selection also work?

Originality: my understanding is that models can process in-context demonstrations better when input output pairs are consecutive. Is there a deeper understanding of why few-shot prompting does better than batch prompting, despite having the same content? Is it simply because training data tends to have sequential (q, a) pairs? If so, the intuition of this paper is to align the prompt more with the data the LLM is trained on. This approach has limited novelty given that it directly combines two existing ideas, as well as standard demonstration selection techniques.

Clarity: Method 1 doesn't seem to generate $a_i$ conditioned on $q_i$. Also, Method 2 appears to generate $a_i$ preceded by $q_1 q_2$, when my understanding is that batch prompting has any $a_i$ preceded by $q_1 q_2 \dots q_n$.

**Questions:**

1. Can you discuss or show results on few-shot prompting
2. Can you discuss or show results on efficiency of the method?

---

> ### Author Response · Authors · 2024-11-15
> **Response for few-shot prompting and efficiency**
>
> Thank your reviews for our work. Some clarity issues will be addressed and the efficiency experiments will be added.
>
> **Motivations:**
> Our main idea is to revisit the detail of the autoregressive inference  of Batch prompting and amplify the demonstration role of batch data. Previous batch prompting works( BatchPrompt: accomplish more with less,  Batch Prompting: Efficient inference..) have claimed that  some previous data in batch may play a role of demonstration. However，there are not enough evidence and explanation on this.
> After we revisit the inference process of batch prompting, we find that the advantage of mutiple data is neglected.  Previous works do not make most of the demonstration role of multiple data in batch prompting, which can be used to improve the performance.
>
> **Novelty**
> The advantage of Autoregressive decoder is that it can generate the smooth text from "left to right".  Previous Batch prompting answers each question in such a way that each question is independent and unrelated, which waste the natural advantage of large models. Our method prompts llm to generate "(question, answer)" pair to address this issue. The "(question, answer)" format of demosntration is significant as "Rethinking the Role of Demonstrations: What Makes In-Context Learning Work https://arxiv.org/pdf/2202.12837”. Also this paper still provide a conclusion  that "Using demonstration is better than not using it" and “only labels” is not working.
>
> We are the first to provide how to construct the demonstrations in the output of batch prompting. And we use the method to cluster the similar data into batches to make most part of results of batch size >=32 better than single data prompting. Our method can provide a more effective batch prompting method with a minimal cost.
>
>
>
> **Extra token cost discussion**
> There are new discussion of the tokens cost of our method.
> The mainly extra token cost comes from the "question" regeneration.  However, we can insert each question instead of the generation of LLM.   As the following  example shows, we add "#" as the mark of the start position of each answer in the prompt. when llm inference "#", which is a mark, we can insert one question from the input question list orderly.  By this means, we save the extra token cost of our method, and the positive effect of demonstration -(question, answer) - generated by llm  can still remain.  Therefore, our method has the same "output tokens" as the conventional batch prompting.
>
> The following is our prompt example to show.
>
>  ```
> You are a professional NLP expert at sentence pair relationship annotation.
> Each data includes a sentence pair, ”Premise” and ”Hypothesis”. Your goal is to classify the sentence pair into two classes as below:
> [class 1]: the given Hypothesis and Premise are logical and following (entailment) to each other.
> [class 0]: the given Hypothesis and Premise are NOT following (entailment) to each other.
> You will be given sentence pairs each time, and the below is the format of sentence
> pairs which will be given:
> ============
> @Sentence pair:
> Premise: xxxxx
> Hypothesis: xxxxx
> ......
> ============
> Below are the outputs you need to generate. ”X” can be ’1’ or ’0’.
> ============
> #Sentence pair:
> Premise: xxxxx
> Hypothesis: xxxxx
> Label: [class X]
> .....
> ============
> ```
> ***embedding role***
> We use embedding model to encoding all the data, and cluster them into batches with the similarities. In this way, we hope to amplify the role of output demonstration in our method. And the result of experiments show the large improvement.  We will explore some other metrics, and some other embedding models, such as "sentence-bert" and "BM-25".
>
>
> ***few-shot prompting***
> We think that conventional few-shot prompting  and our method are totally different. We just want to show the format relation between them. conventional few-shot prompting has only one question to predict, however our "few-shot prompting" is a sequential behavior, each question will work as the demonstraion after being answered. We think they have no necessity to compare their performances.

---

### Note · Authors · 2024-11-12

I have read and agree with the venue's withdrawal policy on behalf of myself and my co-authors.

---

> ### Note · Program_Chairs · 2024-11-12
>
> We approve the reversion of withdrawn submission.

---

### Meta-Review · Area_Chair_qZMA · 2024-12-20

**Metareview:**

This paper studies ways to improve on standard batch prompting (a method that seeks to more efficiently use language models but has limitations) by using a form that is inspired by in-context learning. This basic idea can help improve performance.

The paper’s overall ideas and key observations are quite interesting. There is the nugget of what could be a very interesting paper here. However, as noted by the reviewers, the overall evaluation is very limited. I encourage the authors to continue studying their technique and providing more extensive experimental evidence.

**Additional Comments On Reviewer Discussion:**

Reviewers largely focused on questions around evaluation, and particularly baselines, along with writing clarity. The authors did a solid job of responding to questions around clarity, but did not fully satisfy all of the concerns on evaluation. I largely agreed with the reviewers.

---

### Decision · Program_Chairs · 2025-01-22

Reject